# Antenatal depression: Associations with birth and neonatal outcomes among women attending maternity care in Harare, Zimbabwe

**Malinda Kaiyo-Utete**[1,2]*, **Lisa Langhaug**[2], **Alfred Chingono**[1†], **Jermaine M. Dambi**[2,3,4], **Thulani Magwali**[5], **Claire Henderson**[6☯], **Z. Mike Chirenje**[5,7☯]

1 Department of Primary Health Care Sciences, Mental Health Unit, Faculty of Medicine and Health Sciences, University of Zimbabwe, Harare, Zimbabwe, 2 African Mental Health Research Initiative (AMARI), Research Support Centre, Faculty of Medicine and Health Sciences, University of Zimbabwe, Harare, Zimbabwe, 3 Department of Primary Health Care Sciences, Rehabilitation Unit, Faculty of Medicine and Health Sciences, University of Zimbabwe, Harare, Zimbabwe, 4 The Friendship Bench, Harare, Zimbabwe, 5 Department of Primary Health Care Sciences, Obstetrics and Gynaecology Unit, Faculty of Medicine and Health Sciences, University of Zimbabwe, Harare, Zimbabwe, 6 Department of Health Services and Population Research, King's College London Institute of Psychiatry, Psychology and Neurosciences, London, United Kingdom, 7 Faculty of Medicine and Health Sciences, Clinical Trials Research Centre, University of Zimbabwe, Harare, Zimbabwe

☯ These authors contributed equally to this work.
† Deceased.
* mksutete@gmail.com

**Data Availability Statement:** The data underlying this study are publicly available at https://doi.org/10.5281/zenodo.6105897.

## Abstract

### Introduction

Antenatal depression is highly prevalent and is associated with negative birth and neonatal outcomes. However, the mechanisms and causality behind these associations remain poorly understood as they are varied. Given the variability in whether associations are present, there is need to have context-specific data to understand the complex factors that go into these associations. This study aimed to assess the associations between antenatal depression and birth and neonatal outcomes among women attending maternity care in Harare, Zimbabwe.

### Methods

We followed 354 pregnant women in second or third trimester, attending antenatal care services in two randomly selected clinics in Harare, Zimbabwe. Antenatal depression was assessed using the Structured Clinical Interview for DSM-IV. Birth outcomes included birth weight, gestational age at delivery, mode of delivery, Apgar score, and initiation of breast-feeding within one-hour postdelivery. Neonatal outcomes at six weeks postdelivery included infant's weight, height, illness, feeding methods and maternal postnatal depressive symptoms. The association between antenatal depression and categorical and continuous outcomes were assessed by logistic regression and point-biserial correlation coefficient, respectively. Multivariable logistic regression determined the confounding effects on statistically significant outcomes.

**Funding:** MKU received a PhD fellowship from the DELTAS Africa Initiative [DEL-15-01] through the African Mental Health Research Initiative. The DELTAS Africa Initiative is an independent funding scheme of the African Academy of Sciences (AAS)'s Alliance for Accelerating Excellence in Science in Africa (AESA) and supported by the New Partnership for Africa's Development Planning and Coordinating Agency (NEPAD Agency) with funding from the Wellcome Trust [DEL-15-01] and the UK government. https://amari-africa.org The funders had no role in study design, data collection and analysis, decision to publish or preparation of the manuscript.

**Competing interests:** The authors have declared that no conflicting interests exists

**Abbreviations:** EPDS, Edinburgh Postnatal Depression Scale; SCID-IV, Structured Clinical Interview for Diagnostic and Statistical Manual version IV.

## Results

Prevalence of antenatal depression was 23.7%. It was associated with low birthweight [AOR = 2.30 (95% CI: 1.08–4.90)], exclusive breastfeeding [AOR = 0.42 (95%CI: 0.25–0.73)] and postnatal depressive symptoms [AOR = 4.99 (95%CI: 2.81–8.85)], but not with any other birth or neonatal outcomes measured.

## Conclusions

The prevalence of antenatal depression in this sample is high with significant associations demonstrated for birth weight, maternal postnatal depressive symptoms and infant feeding methods Effective management of antenatal depression is thus crucial to the promotion of maternal and child health.

## Introduction

Antenatal depression affects pregnant women from conception through to delivery and is highly prevalent [1]. About 10% of all pregnant women globally become depressed antenatally, and more in low- and middle-income countries (LMICs) [1, 2]. Recent studies in urban Zimbabwe found a prevalence of antenatal depression of 23.5% among pregnant women [3], and a prevalence of 39.4% among HIV+ pregnant women [4]. Antenatal depression is often underdiagnosed, and therefore undertreated [5]. One possible factor in its treatment gap is that depression symptoms are mistaken for normal physiological changes in pregnancy [5]. Its undertreatment has been associated with negative effects on the mother and her developing foetus [6, 7].

Poor health seeking behaviours [8, 9], risky sexual behaviours [10] and illicit substance use [11] as well as lack of self-motivation to follow prescribed and recommended antenatal care regimens [12] among depressed pregnant women may contribute to unfavourable pregnancy outcomes. Previous studies have shown a strong association between antenatal depression and low birth weight [13, 14], preterm birth [14]. These findings have been confirmed by recent systematic reviews [15–17]. Antenatal depression has been associated with poor breastfeeding practices [18]. This is noteworthy given that breastfeeding is considered the most effective public health intervention for infant mortality and morbidity [19]. Postnatal depression is often a sequela of untreated antenatal depression, with almost 50% of women who are diagnosed with postnatal depression having had signs of depression in pregnancy [20]. Postnatal depression is well-studied and its profound negative effects on the infant's development are well documented [21, 22].

Some studies, however, have shown that there are no statistically significant associations between antenatal depression and adverse birth and neonatal outcomes [23–26]. Most regional studies showing mixed results on the associations between antenatal depression and birth/neonatal outcomes have been nested within larger trials that were not designed to answer this specific study question [27, 28]. A study of a large Ghanaian cohort of pregnant women aimed to reduce maternal mortality through weekly vitamin A supplementation and to assess impact of home visits by community volunteers. These investigators showed that there was no association between antenatal depression and neonatal deaths, still births, low birth weight, delayed initiation of breastfeeding, non-exclusive breastfeeding in the neonatal period, delivering at a health facility, or optimal antenatal care attendance [27]. However, the same study showed

that there was an increased risk of prolonged labour, peripartum and postpartum complications, severe new-born illness and risky behaviours such as non-use of mosquito nets during pregnancy among women with antenatal depression [27]. In a study in Malawi, nested within a trial of nutritional supplementation, antenatal depression was also not associated with birth weight, gestational age, neonatal length-age-Z-score, or head-circumference-Z-score but was inversely associated with mid-upper arm circumference [28]. In both studies, the inconsistencies could be due to methodological differences. For example, both Ghanaian and Malawian studies used screening tools that have been shown to overestimate the prevalence of antenatal depression compared to diagnostic tools [29]. Nutritional supplementation in these studies could have provided a beneficial effect on maternal health including reducing levels of antenatal depression [28]. Given their limitations, the studies did not resolve the conflicting findings regarding the association between antenatal depression and prematurity and/or low birth weight and other obstetric outcomes in sub-Saharan Africa.

Due to these conflicting reports on the effects of antenatal depression, and paucity of evidence on the associations of antenatal depression in the Zimbabwean context, which has a high prevalence of antenatal depression [3, 4], we sought to examine these associations among pregnant women attending maternity services at Harare clinics. If antenatal depression is associated with poor birth and neonatal outcomes and maternal postnatal depressive symptoms, it would emphasize the importance of screening for it as part of routine prenatal care

## Methods

This explorative prospective cohort study followed a validation study of two commonly used depression screening tools (Edinburgh Postnatal Depression Scale and Patient Health Questionnaire-9) conducted at two randomly selected Harare City Council clinics. Data collection occurred from January to December 2018. There are 12 polyclinics under the Harare City Council Directorate which offer primary health care services to people of low socioeconomic status in the city [30]. The researcher wrote each clinic's name on a piece of paper and placed the 12 pieces of paper in a basket. She asked a colleague to randomly pick two pieces of paper out of the basket. The clinics picked were thus selected as study sites for data collection. The clinics are staffed by midwives and nurses, with a medical doctor attending to severe chronic conditions monthly [30]. At the time of the study, every pregnant woman paid an equivalent of US$25.00 for maternity services to cover care from the time of initial intake to six weeks post-delivery.

Pregnant women in their second or third trimester, aged ≥16 years, able to communicate in English or Shona, and registered for antenatal care at one of the two selected Harare polyclinics were included in the study. Those with severe mental illnesses such as schizophrenia and bipolar disorder, not able to provide written informed consent and those who were unwilling to be followed until six weeks post-delivery were excluded. Assuming a 39% prevalence of antenatal depression [31], we calculated that a sample of 366 women was needed to achieve a 5% precision at 95% confidence intervals.

### Measures

**Exposure.** Antenatal depression was assessed using the Structured Clinical Interview for the Diagnostic and Statistical Manual fourth version (SCID-IV) [32]. SCID-IV is administered by trained mental health professionals [32]. It has three categories for the diagnosis of depression: "no depression", "minor depression" and "major depression". For this analysis, depression status in pregnancy was further dichotomised into "depressed", which included those who met criteria for "minor depression" and "major depression", or "not depressed", for those

who had no depression. The tool had been translated and previously used in Zimbabwe to assess postnatal depression [3, 33]).

## Outcome measures

**a. Birth outcomes.** We collected the infant's birthweight, gestational age at birth, mode of delivery, Apgar score at 5 minutes, and whether breastfeeding was initiated within one-hour postdelivery from maternal and/or infant's medical booklets. Infant birthweight was measured in grams upon delivery in the labour room by the attending midwife, or as soon as the mother presented at the health facility within 24 hours postdelivery for those who delivered before arriving at the facility. Birthweight was categorised into low birthweight (LBW), for birthweight <2,500g, and normal birthweight (NBW) for birthweight ≥2,500g. Gestational age at birth was estimated as the time between the women's first day of last normal menstrual period as reported by the woman and infant's delivery. It was then categorized into 1) preterm birth (PTB), which takes place before 37 completed weeks of gestation, and 2) term birth, which takes place at 37 weeks of gestation or more. Mode of delivery was either "assisted delivery" which included those who delivered by caesarean section, vacuum extraction, forceps, or episiotomy, or "normal delivery" for those who had a spontaneous vaginal delivery. Apgar score was also dichotomised into "low", a score of <7 and "normal", a score ≥7. Mothers either initiated breastfeeding within 1-hr postdelivery ("Yes") or did not ("No").

**b. Neonatal outcomes.** At the six weeks postdelivery clinic visit, we collected the infant's weight, height, whether the infant had been ill as recorded in the medical records prior to the visit, and the infant's feeding methods were recorded. Infant's weight and height were measured routinely by the attending midwife or nurse and recorded in the baby's medical booklet. Research team members checked the infant's medical records to assess whether the infant had had a clinic visit for diarrhoea, respiratory infection, or any other illness in the previous six weeks. Infant's feeding methods were assessed by asking the mother's self-report and categorized what she was feeding her infant. Infant's feeding methods were either "exclusive breastfeeding" if the baby was feeding on breast milk only, or "other" if breastmilk was supplemented by other foods such as cereal, porridge or juices.

**c. Maternal outcomes.** Maternal postnatal depressive symptoms at six weeks postdelivery post-delivery were assessed using the Shona-version of the Edinburgh Postnatal Depression Scale (EPDS). The 10-item screening tool, originally developed in the UK for postnatal depression [34], is the most widely used and validated maternal mental health screening tool [35–37]. It was validated among Zimbabwean postnatal women with a cut-off score of 11/12 which yielded a sensitivity of 88%, specificity of 87% [38]. The positive predictive value was 74%, a negative predictive value was 94%, and an area under the curve was 0.82 [38].

*Confounding measures*. A priori reproductive and socio-demographic risk factors for antenatal depression and for adverse birth and neonatal outcomes were measured as potential confounders. These included maternal age at baseline, parity, previous pregnancy complications, chronic illnesses diagnosed during or prior to current pregnancy, maternal HIV status, maternal body mass index at baseline and history of intimate partner violence in the past three months. These variables were chosen based on evidence from previously published research [13, 39–41].

## Data collection procedures

Baseline data, which included participants' sociodemographic characteristics, medical and obstetric history, and psychosocial issues, were collected from January to April 2018 by the research team as reported in a previous manuscript [3]. Research assistants administered the

study questionnaire while the study psychiatrists administered the SCID-IV. Each participant's expected date of delivery (EDD) was recorded, and then telephonically made appointments with participants to assess their birth outcomes. The scheduled appointments were planned to coincide with her 3- or 7-day postdelivery clinic visit. If, at the time of the phone call, the woman had not yet given birth, a follow-up phone call was made once a week.

On the day of the appointment, we met the participant after her scheduled clinic visit. The investigator then checked both the mother's and infant's medical records and recorded the birth outcomes on an android tablet. After the meeting, the investigator then made another appointment with the mother to coincide with her scheduled six-week post-delivery clinic visit.

At the six-week post-delivery visit, the investigator collected the infant's weight, height, whether the infant had been ill since birth and how the baby was feeding from the infant's medical records. Additionally, the Shona-version of the EPDS was administered to assess the mother's depressive symptoms six weeks post-delivery.

All data were collected and recorded on an Android tablet, and data were uploaded onto an online data server.

## Statistical analysis

Descriptive statistics such as frequencies and means with standard deviations were used to describe the study sample's demographic characteristics, antenatal depression status, birth, and neonatal outcomes. Logistic regression analysis was used to assess the strength and direction of the association between antenatal depression and categorical outcomes: birthweight, gestational age at birth, Apgar scores, infant illnesses, feeding methods and maternal depressive symptoms at six weeks postdelivery. Point-biserial correlation coefficient was used to measure the strength and direction of the association between antenatal depression and continuous outcomes such as infant's weight and height at six weeks postdelivery. Multivariate regression analysis was carried out to assess the effects of confounders. The strength of the association of antenatal depression and birth/neonatal outcomes was expressed as odds ratios (OR) with 95% confidence interval. Multicollinearity was assessed using correlation coefficients and none of the covariates were found to be correlated. All tests were two-tailed, and significance level was at $p<0.05$. All statistical analysis was done in STATA version 14 (2015).

## Ethical considerations

The study was approved by the Joint Research Ethics Committee (JREC) for the University of Zimbabwe, College of Health Sciences and The Parirenyatwa Group of Hospitals (JREC/158/17) and the Medical Research Council of Zimbabwe (MRCZ/A/2209). Permission to carry out the study at the study clinics was granted by the Harare City Health Directorate and the clinics' management. Written informed consent was obtained from the prospective participants. Pregnant women aged ≤16 years were considered emancipated adults; therefore, no parental consent was obtained. Participants who met the diagnostic criteria for depression were referred for further management to the mental health nurse or the Friendship Bench, a problem-solving psychotherapy administered by lay health workers.

## Results

Three hundred and seventy-five pregnant women were recruited into the study, and 354 were followed up at birth. Twenty-one women were lost to follow up.

## Sociodemographic characteristics

Table 1 shows the participants' sociodemographic characteristics. The largest proportion of the participants were aged below 25 years (n = 161, 45.5%). Most had attained a secondary school level education (n = 297, 83.9%). Sixty-two percent (n = 221) were primipara and above. Of the 221 primipara and above, 76 (21.5%) had previous birth complications. About a quarter of the participants (n = 88) were HIV-positive.

## Depression status in pregnancy

Eighty-four participants (23.7%) had antenatal depression according to the SCID-IV. Twenty-nine (34.5%) pregnant women who were depressed had major depression compared to 55/84 (65.5%) who had major depression.

## Birth and neonatal outcomes

As shown in Table 2, of the 354 women who were seen three to seven days postnatally, three (0.84%) had had early neonatal death, 39 (11.0%) had low birthweight babies, 52 (14.7%) had preterm deliveries, 51 (14.4%) had assisted deliveries, 13 (3.7%) had babies with low Apgar score and 22 (6.2%) had not initiated breastfeeding within one hour of delivery.

At the six weeks postdelivery assessment, eight (2.8%) infants had died in the neonatal period. The infant's mean weight was 5021.5g [±715.5 (95%CI: 4945.4–5097.6)]. Their mean length 54.7cm [±2.7 (95%CI: 54.4–55.0)]. Of the 343 participants whose babies were alive, 110 (31.8%) were not breastfeeding exclusively, 76 (21.7%) participants' infants had been ill as recorded in their medical records and 102 (28.8%) participants had depressive symptoms based on the EPDS at the six-week postdelivery visit; of these 44 (43.1%) were depressed in pregnancy. Of the 76 (21.7%) infants who had been ill, 47 (13.4%) had acute respiratory infection, 21 (6.0%) had diarrhoeal infection and 8 (2.3%) had other illnesses including seizures, conjunctivitis, and jaundice.

## Associations between antenatal depression and adverse birth and neonatal outcomes

Table 3 shows the unadjusted associations between antenatal depression and birth and neonatal outcomes. There was no statistically significant association between antenatal depression and infant's weight ($r_{po}$ = -0.059, p = 0.278) or height ($r_{po}$ = -0.067, p = 0.218) at six weeks postdelivery.

Table 4 shows the results from multivariate regression analysis. The associations between antenatal depression and each of low birthweight [Adjusted OR = 2.30 (95%CI: 1.08–4.90), p = 0.03], exclusive breastfeeding [Adjusted OR = 0.42 (95%CI:0.25–0.73), p = 0.002] and maternal depressive symptoms at six weeks [Adjusted OR = 4.99 (95%CI:2.81–8.85), p<0.001] was statistically significant after controlling for maternal age, parity, previous birth complications, chronic illnesses, HIV status, intimate partner violence and maternal body mass index (BMI).

## Discussion

The study aimed to determine associations between antenatal depression and birth and neonatal outcomes. To our knowledge, this is the first study in Zimbabwe to examine these associations. Study results showed that almost a quarter of pregnant women attending antenatal clinics in Harare, Zimbabwe were depressed, and almost one-third of women had depressive symptoms at six weeks postdelivery. The point estimates for the odds ratios suggest a two-fold

**Table 1. Participants' sociodemographic characteristics (N = 354).**

| Characteristic | Frequency (%) |
|---|---:|
| Maternal age in years | |
| Below 25 | 161 (45.5%) |
| Between 25 and 29 | 100 (28.2%) |
| 30 and above | 93 (26.3%) |
| Residence | |
| High density | 292 (82.5%) |
| Other | 62 (17.5%) |
| Occupation | |
| Not employed | 218 (61.6%) |
| Employed | 136 (38.4%) |
| Educational level | |
| Primary level | 26 (7.3%) |
| Secondary school level | 297 (83.9%) |
| Tertiary level | 31 (8.8%) |
| Importance of religion | |
| Very important | 330 (93.2%) |
| Not important | 24 (6.8%) |
| Parity | |
| Nullipara | 133 (37.6%) |
| Primipara and above | 221 (62.4%) |
| Intended pregnancy | |
| Yes | 265 (74.9%) |
| No | 89 (25.1%) |
| History of obstetric complications | |
| No | 145 (41.0%) |
| Yes | 76 (21.5%) |
| No previous birth | 133 (37.6%) |
| Chronic illness diagnosed in current pregnancy | |
| Yes | 29 (8.2%) |
| No | 325 (91.8%) |
| Chronic illness diagnosed prior to current pregnancy | |
| Yes | 62 (17.5%) |
| No | 292 (82.5%) |
| HIV status | |
| Positive | 88 (24.9%) |
| Negative | 266 (74.1%) |
| Marital status | |
| Married/cohabitating | 280 (79.1%) |
| Other | 74 (20.9%) |
| Negative life event in the past one year | |
| No | 185 (52.3%) |
| Yes | 169 (47.7%) |
| Have someone to talk to when overwhelmed | |
| No | 125 (35.3%) |
| Yes | 229 (64.7%) |
| History of intimate partner violence in the past 3 months | |
| No | 238 (67.2%) |

*(Continued)*

**Table 1.** (Continued)

| Characteristic | Frequency (%) |
|---|---|
| Yes | 116 (32.8%) |
| Alcohol use in pregnancy | |
| No | 319 (90.1%) |
| Yes | 35 (9.9%) |
| Maternal BMI | |
| Below to normal BMI | 180 (50.1%) |
| Above normal | 174 (49.1%) |
| Depression in pregnancy | |
| No | 270 (76.3%) |
| Yes | 84 (23.7%) |

increased risk of low birth weight and four-fold increased risk of maternal postnatal depressive symptoms for women with antenatal depression. Mothers who had antenatal depression were less likely to exclusively breastfeed their babies at six weeks postdelivery.

**Table 2. Birth and neonatal outcomes (N = 354).**

| Outcome | Total (N = 354) |
|---|---|
| Birth weight at delivery | |
| Low Birth weight | 39 (11.0%) |
| Normal/above birthweight | 315 (89.0%) |
| Mode of Delivery | |
| Assisted delivery | 51 (14.4%) |
| Normal delivery | 303 (85.6%) |
| Gestational age at delivery | |
| Preterm delivery | 52 (14.7%) |
| Term delivery | 302 (85.3%) |
| Apgar score at 5mins post-delivery | |
| Subnormal (<7) | 13 (3.7%) |
| Normal | 341 (96.3%) |
| Initiating breastfeeding within 1hr postdelivery | |
| Yes | 22 (6.2%) |
| No | 332 (93.8%) |
| Infant height at 6 weeks post-delivery [#] | 54.7±2.7 (95%CI: 54.4–55.0) |
| Infant weight at 6 weeks post-delivery [#] | 5021.5±715.5 (95%CI: 4945.4–5097.6) |
| Infant Feeding Methods at 6 weeks post-delivery[#] | |
| Non-exclusive breastfeeding | 110 (31.8%) |
| Exclusive breastfeeding | 232 (67.1%) |
| Infant illness in the first six weeks of life[##] | |
| Yes | 76 (21.7%) |
| No | 275 (78.3%) |
| Postnatal depressive symptoms at six weeks post-delivery (EPDS) | |
| ≥11 | 102 (28.8%) |
| <11 | 252 (71.2%) |

[#]N = 343
[##]N = 351

**Table 3. Associations of birth and neonatal outcomes with antenatal depression (N = 354).**

| Outcome | Depressed (N = 84) | Not Depressed (N = 270) | Unadjusted | |
|---|---|---|---|---|
| | | | Odds Ratio (95%CI) | p-value |
| Birth weight at delivery | | | | |
| Normal/above Birthweight | 70 (19.8%) | 245 (69.2%) | | |
| Low birthweight | 14(3.9%) | 25 (7.1%) | 1.96 (0.97–3.97) | 0.06 |
| Mode of Delivery | | | | |
| Normal delivery | 73 (20.6%) | 230 (64.9%) | | |
| Assisted delivery | 11 (3.1%) | 40 (11.3%) | 0.87 (0.42–1.77) | 0.70 |
| Gestational age at delivery | | | | |
| Term delivery | 73 (20.3%) | 229 (64.7%) | | |
| Preterm delivery | 11 (3.1%) | 41 (11.6%) | 0.84 (0.41–1.72) | 0.64 |
| Apgar score at 5mins post-delivery | | | | |
| Normal | 81 (22.9%) | 260 (73.4%) | | |
| Subnormal | 3 (0.8%) | 10 (2.8%) | 0.96 (0.26–3.58) | 0.95 |
| Initiating breastfeeding within 1hr postdelivery | | | | |
| Yes | 76 (21.5%) | 256 (72.3%) | | |
| No | 8 (2.3%) | 14 (3.9%) | 1.92 (0.78–4.76) | 0.16 |
| Infant Feeding Methods at 6 weeks post-delivery* | | | | |
| Non-exclusive feeding | 39 (11.39%) | 72 (20.99%) | | |
| Exclusive breastfeeding | 42 (12.24%) | 190 (55.39%) | 0.42 (0.25–0.70) | 0.001* |
| Infant illness in the first six weeks of life ## | | | | |
| No | 65 (18.5%) | 210 (59.8%) | | |
| Yes | 17 (4.8%) | 59 (16.8%) | 0.93 (0.51–1.71) | 0.82 |
| Postnatal depressive symptoms at six weeks post-delivery | | | | |
| Not Depressed | 40 (11.3%) | 212 (59.9%) | | |
| Depressed | 44 (12.4%) | 58 (16.4%) | 4.02 (2.39–6.75) | ≤0.001* |

#N = 343

##N = 351*, statistically significant at p<0.05

The prevalence of low birth weight in the study sample was 11.0%, which was lower than that found in a previous Zimbabwean study of 16.7% [42]. The statistically significant association between antenatal depression and low birth weight was consistent with a meta-analysis carried out of 11 studies that showed a relative risk ranging from 1.02–4.75 [24] of pregnant women with depression delivering babies with low birth weight. The result is also consistent with other recent systematic reviews [15–17]. One possible explanation for this association is that mothers with depression have reduced cognitive function, which may lead to poor maternal nutrition, poor health seeking behaviours and increased risk behaviours such as alcohol use and smoking which may ultimately affect the foetal intrauterine growth [11]. This result has important clinical significance. Low birth weight infants are at a higher risk of neonatal morbidity and mortality [43, 44]. It is imperative therefore to address every aspect, including maternal mental health, that may be associated with low birth weight to reduce infant mortality and morbidity.

The study results showed that 22 (6.2%) participants did not initiate breastfeeding within one-hour post-delivery as is recommended by the WHO [23], which was lower than previously recorded in Zimbabwe [45]. This could be attributed to any number of reasons, such as the individual clinic context (if initiation of breastfeeding within hour was encouraged or not), recovering from general anaesthesia following a caesarean section or having delivered infants

**Table 4. Multivariate analysis for the associations between antenatal depression and adverse birth and neonatal outcomes.**

| Predictor | Low birth weight | | Infant illness within 6 weeks postdelivery | | Maternal postnatal depressive symptoms | |
|---|---|---|---|---|---|---|
| | Adjusted OR (95% CI) | p-value | Adjusted OR (95% CI) | p-value | Adjusted OR (95% CI) | p-value |
| Antenatal depression | | | | | | |
| Not depressed | 1 | | 1 | | 1 | |
| Depressed | 2.30 (1.08–4.90) | 0.03* | 0.89 (0.45–1.75) | 0.73 | 4.99 (2.81–8.85) | <0.0001* |
| Maternal age group | | | | | | |
| Less than 25 | 1 | | 1 | | 1 | |
| 25 to below 30 | 0.67 (0.26–1.74) | 0.41 | 0.67 (0.33–1.36) | 0.27 | 1.86 (0.97–3.56) | 0.06* |
| 30 and above | 1.15 (0.47–2.85) | 0.75 | 0.37 (0.68–.083) | 0.02* | 1.72 (0.85–3.48) | 0.129 |
| Parity | | | | | | |
| Nullipara | 1 | | 1 | | 1 | |
| Multipara | 1.19 (0.49–2.90) | 0.71 | 1.99 (0.98–4.01) | 0.06 | 0.58 (0.31–1.11) | 0.099 |
| Previous birth complications | | | | | | |
| No | 1 | | 1 | | 1 | |
| Yes | 1.04 (0.42–2.51) | 0.93 | 0.84 (0.41–1.74) | 0.66 | 0.90 (0.46–1.76) | 0.76 |
| No previous pregnancy | 1 (omitted) | | 1 | | 1 (omitted) | |
| Chronic illnesses diagnosed during current pregnancy | | | | | | |
| No | 1 | | 1 | | 1 | |
| Yes | 0.43 (0.09–2.12) | 0.30 | 0.84 (0.32–2.23) | 0.73 | 0.53 (0.20–1.44) | 0.21 |
| Chronic illness diagnosed prior to this current pregnancy | | | | | | |
| No | 1 | | 1 | | 1 | |
| Yes | 1.23 (0.48–3.16) | 0.67 | 1.71 (0.86–3.39) | 0.13 | 0.88 (0.42–1.78) | 0.71 |
| HIV status | | | | | | |
| Negative | 1 | | 1 | | 1 | |
| Positive | 0.96 (0.41–2.25) | 0.93 | 4.38 (2.38–8.04) | <0.001* | 1.07 (0.58–1.95) | 0.83 |
| History of intimate partner violence | | | | | | |
| No | 1 | | 1 | | 1 | |
| Yes | 0.66 (0.30–1.43) | 0.29 | 1.79 (1.01–3.18) | 0.05* | 0.79 (0.46–1.37) | 0.46 |
| BMI | | | | | | |
| Below to normal | 1 | | 1 | | 1 | |
| Above normal | 1.00 (0.50–2.03) | 0.98 | 1.01 (0.57–1.79) | 0.96 | 1.37 (0.83–2.27) | 0.23 |

*statistically significant at $p<0.05$

with very low Apgar scores who needed immediate medical attention [45]. Although women who were depressed in pregnancy were twice as likely not to initiate breast feeding early, this association was not statistically significant. This result is contrary to a recent systematic review on the association between antenatal depression and delayed initiation of breastfeeding which showed a strong, positive association between the two [18]. One-third of the participants were not exclusively breastfeeding their babies at six-week postnatal assessment point. This result is consistent with previously published studies from Zimbabwe which showed that about 66.7% of mothers exclusively breastfeed their infants [46–48]. The low uptake of exclusive breastfeeding could be due to cultural barriers, such as elder women's interference in the caring of the infant, that young mothers face when trying to implement this [46, 47, 49]. In our study, mothers with antenatal depression were less likely to exclusively breastfeed their infants at six weeks postdelivery, although this association was quite small. The negative association between antenatal depression and exclusive breastfeeding is consistent with results from previous studies

[18, 50–55]. Women who are already struggling with depression would have even greater difficulty pushing back against these cultural barriers hence the strong association. However, regionally, it has been shown that depression does not affect intention and initiation of breastfeeding or exclusive breastfeeding [27, 56]. Since breastfeeding, particularly exclusive breastfeeding, are the most effective public health interventions for reducing infant mortality and morbidity, any known factor, including antenatal depression, should be addressed to improve uptake of breastfeeding.

The prevalence of postnatal depressive symptoms among the study sample was 28.8%. The result is consistent with previous studies in Zimbabwe that have shown a high prevalence of postnatal depression ranging from 21% to 33% [33, 57, 58]. Our study also showed that pregnant women who were depressed in pregnancy were four-times more likely to have depressive symptoms at six weeks post-delivery, which is in line with prior studies on the association between prenatal and postnatal depression [14, 20]. Research has shown that postnatal depression has negative effects on the infant's growth and development [22]. It is therefore recommended that pregnant women be screened for antenatal depression, and positively screened cases be managed effectively, to reduce the odds of postnatal depression.

Multivariate analysis also showed that some birth and neonatal outcomes, such as low birth weight, assisted delivery, initiation of breastfeeding with one-hour postdelivery and infant illness at six weeks postdelivery were associated with other factors such as maternal age, parity, maternal HIV status and history of intimate partner violence within the past three months. This showed how complex and intersecting the factors that affect birth and neonatal outcomes are, other than antenatal depression.

In this study, antenatal depression was statistically associated with low birth weight, infant feeding methods and postnatal depressive symptoms, but not with preterm delivery, assisted deliveries, initiating breastfeeding within one-hour post-delivery, low Apgar scores, infant's weight, and height at six weeks postdelivery, and neonatal infant illness. These results were like those arising from other studies in sub-Saharan countries [27, 28]. Lack of a statistically significant association between antenatal depression and infant illness demonstrated in our study is surprising. As antenatal depression can affect a mother's ability to follow recommended medical regimens [6, 10], it follows that her infant could be at risk of ill-health, but we did not find this to be the case. This result contradicts findings in the Ghana [27] which showed a strong association between antenatal depression and serious infant illness in the postnatal period. The lack of statistical significance between depression and certain outcomes, such as infant illness, may be because most participants were diagnosed with minor depression, some may have received treatment and others may have comorbid anxiety or other mental disorder which may have affected outcomes [58, 59].

The study had some limitations. Firstly, antenatal depression was assessed once, the participant might have not been depressed at the time of data collection but later developed depression or vice versa. It would also be preferable to follow pregnant women starting in the first trimester, which is considered the most critical stage in foetal development. However, recruiting women in the first trimester may not be feasible in the Zimbabwean setting, as most pregnant women present for antenatal care in the second and third trimester [60, 61]. We also categorised major depression and minor depression as depressed; however, severity of depressive symptoms is likely to affect pregnancy outcomes differently. Although the pregnant women who had SCID-diagnosed minor/major depression were referred to the Friendship Bench [62] or mental health nurse for further management, information on management outcomes was not available; the fact that women may have received treatment for antenatal depression could explain some of the lack of association between antenatal depression and many birth and neonatal outcomes. A larger cohort study following pregnant women from the

first trimester with depression being assessed in all trimesters at multiple points is recommended. Another limitation is that the study showed associations, and not causality.

Despite the above-stated limitations, the study has several strengths. The prospective cohort study used a diagnostic tool, SCID-IV, to assess antenatal depression unlike most studies which used screening tools such as the PHQ-9 [27], EPDS [14, 55, 63], SRQ-20 [25, 28] and BDI-II [26]. Screening tools have been shown to overestimate prevalence of depressive symptoms [63]. To reduce recall and/or measurement bias, the participants' responses to study outcomes were collected directly from the maternal and infant medical records. The study also had a low attrition rate which led to collection of comprehensive data.

## Conclusion

The association between antenatal depression and low birth weight, non-exclusive breastfeeding at six weeks postdelivery and maternal postnatal depressive symptoms has major clinical significance in maternal and child health care. Low birth weight is the highest cause of infant mortality and morbidity hence every possible contributing factor needs to be addressed to promote maternal and child health. Accordingly, there is great need for more research to identify whether screening for and subsequently treating antenatal depression can improve these outcomes; this data is needed to support universal screening for antenatal depression. In our low-income settings, task-shifted mental health interventions can potentially be used to address antenatal depression in primary or maternal health settings [61] could and thereby reduce its negative effects of on birth and neonatal outcomes.

## Acknowledgments

We thank the women who participated in the study for their time, the research team who collected data and the study clinic managers and their staff for assisting with logistics during data collection. We are also grateful to the Mildred Nemaramba, who assisted in the analysis of the data and Helen Jack who assisted with editing the manuscript.

## Author Contributions

**Conceptualization:** Malinda Kaiyo-Utete, Lisa Langhaug, Alfred Chingono, Jermaine M. Dambi.

**Data curation:** Malinda Kaiyo-Utete.

**Formal analysis:** Malinda Kaiyo-Utete, Jermaine M. Dambi, Claire Henderson.

**Funding acquisition:** Malinda Kaiyo-Utete.

**Investigation:** Malinda Kaiyo-Utete, Z. Mike Chirenje.

**Methodology:** Malinda Kaiyo-Utete, Lisa Langhaug, Alfred Chingono, Jermaine M. Dambi, Claire Henderson.

**Project administration:** Malinda Kaiyo-Utete.

**Resources:** Malinda Kaiyo-Utete.

**Supervision:** Lisa Langhaug, Alfred Chingono, Thulani Magwali, Claire Henderson, Z. Mike Chirenje.

**Validation:** Lisa Langhaug, Thulani Magwali, Claire Henderson, Z. Mike Chirenje.

**Writing – original draft:** Malinda Kaiyo-Utete, Lisa Langhaug, Alfred Chingono, Claire Henderson.

**Writing – review & editing:** Malinda Kaiyo-Utete, Lisa Langhaug, Alfred Chingono, Jermaine M. Dambi, Thulani Magwali, Claire Henderson, Z. Mike Chirenje.

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
