## [Decision Letter · Decision Letter 0]

4 Dec 2021

PONE-D-21-18898Antenatal depression: Associations with birth and neonatal outcomes among women attending maternity care in Harare, ZimbabwePLOS ONE

Dear Dr. Kaiyo-Utete,

Thank you for submitting your manuscript to PLOS ONE. After careful consideration, we feel that it has merit but does not fully meet PLOS ONE’s publication criteria as it currently stands. Therefore, we invite you to submit a revised version of the manuscript that addresses the points raised during the review process.

 The authors should pay particular attention to their methodological section to clarify important points that the reviewers raised, and in particular to the data analysis section. These changes may then re-inform the Results. 

We look forward to receiving your revised manuscript.

Kind regards,

Michael Wells

Academic Editor

PLOS ONE

Additional Editor Comments:

Both reviewers see important merit in the submitted manuscript, but they both raise important methodological and data analysis inquiries that need to be addressed.

Reviewers' comments:

Reviewer's Responses to Questions

**Comments to the Author**

1. Is the manuscript technically sound, and do the data support the conclusions?

Reviewer #1: Yes

Reviewer #2: Partly

2. Has the statistical analysis been performed appropriately and rigorously? 

Reviewer #1: No

Reviewer #2: Yes

3. Have the authors made all data underlying the findings in their manuscript fully available?

Reviewer #1: No

Reviewer #2: No

4. Is the manuscript presented in an intelligible fashion and written in standard English?

Reviewer #1: Yes

Reviewer #2: Yes

5. Review Comments to the Author

Reviewer #1: This is a very interesting paper aiming to examine the prevalence of antenatal depression and its associations with birth and neonatal outcomes in Zimbabwe. The manuscript is clear, well written and discusses about an important topic. However, there are several pieces of information missing and several points that require improvement, as described below:

Abstract:

Please make it already very clear in the abstract what are the main aims of the study.

Introduction

Throughout the text, there are some typos and grammar mistakes. Please proofread and review the paper carefully.

The introduction is very well-written, but I am missing a more thorough review of the studies conducted in Africa, but specifically in Zimbabwe.

Also, some citations are missing (e.g. “Its under treatment has been associated with negative effects

on the mother and her developing foetus.“;“ paucity of evidence on the associations of antenatal depression in the Zimbabwean context, which has a high prevalence of antenatal depression“). The reader has no clue if certain statements refer to the African or to the global context or if they are based on assumptions or evidence. Please cite accordingly.

Methods

Was the validation of the PHQ-9 and EPDS done with this same population or just in the same health centres?

Please describe in details how the randomization process for the maternity clinics was conducted.

Did the participants provide written consent? If yes, please clearly state it.

Please describe in details the power analysis conducted to decide on the number of participants needed.

The prenatal depression diagnostics was done with the SCID interview. How did you account for physical symptoms relevant to the depression diagnosis which are normal in pregnancy?

What is the relevance of the variable “initiated breastfeed within 1h post-delivery” for the research question? How is this associated with prenatal depression given that many times, depending on the maternity care centre procedures, this initiation or lack of, is not a maternal choice?

Did the authors account for current pregnancy complications or only for past complications? If not current, why not?

Please cite the studies supporting the choice for the confounding variables. I am unsure if they belong to the global or regional level.

Authors mentioned that participants were recorded as depressed or non-depressed. But from those scored as depressed, what is the percentage/proportion of minor and major depression? Please add this to the text.

Authors mentioned that the depressed participants were referred to the mental health nurse or psychotherapy. When did participants started with this treatment? What kind of treatment was given? How has this impacted mental health measurements postpartum? You need to adjust for this important variable in the analysis.

Results

Were there any differences between the participants who remained until the end of the study and the N=21 who were lost in the follow up?

Participants can be either primipara or multiparas, not both. Please correct it in the text.

Table1: the authors mentioned that covariates were chosen based on previous literature and evidence, but aren’t religion and social support quite important in the regional context? Also, were the randomly selected clinics settled in a rural or urban region? This should be acknowledged, given that if they are in an urban setting, women attending it might not be comparable to those in a rural setting with less resources to attend and pay for the maternity care.

Please state the percentage of participants diagnosed with minor and major depression in the sample in Table 1.

Although the authors clearly described depression levels and infant characteristics in the postpartum phase, it would be interesting to see a table with the outcomes from babies whose mothers were antenatally depressed. It is unclear, especially considering prenatally depressed mothers were offered and potentially received treatment during pregnancy, how many of those were still depressed in the postpartum and the potential effects/improvement on baby outcomes for those receiving therapy.

Table 2: It would be important to add the percentage of babies not being breastfed at all (e.g. only formula feeding).

Discussion

In the second paragraph, authors justify the association between antenatal depression and LBW risk behaviours. However, it is important to contextualize it to your sample. Based on Table 1, the participants have a low percentage of risk behaviours. Maybe you could justify it if the depressed moms have a higher rate of risky behaviours upon comparison with the non-depressed one. But this needs to be described in text or tables.

Authors also mentioned cultural barriers associated with breastfeeding. What are those in Zimbabwe? It would be important to contextualize it in the region where the study was conducted as well as with the sample here studied.

Please make sure to make it clear that the association between prenatal depression and exclusive breast was quite small and how relevant is it.

Did the authors considered that the lack of statistical significance between depression and certain outcomes may be due to the fact that most participants were diagnosed with minor depression? Since this is not clearly described in text, the reader has no clue if this is a possibility. Please make it clear.

Reviewer #2: In this original article, the Authors aim to determine the association between antenatal depression during the second or third trimester and birth and neonatal outcomes assessed at 6 weeks post-partum in a cohort of 354 Zimbabwean women. They found that a diagnosis of major and minor antenatal depression was associated with low birth weight in the new-borns, with a lower rate of exclusive breastfeeding and with higher rate of depressive symptoms beyond the EPDS cut-off.

The study is interesting because to date only few prospective studies evaluated the impact of prenatal depression in sub-Saharian countries. Some methodological issues, hovewer, lower my enthusiasm for the manuscript. Specifically the authors considered the antenatal depression as the main predictor of ten different outcomes. In a second step of the analyses they controlled for eight different covariates. Moreover they dichotomised most of the variable with the risk of losing some important information.

If possible I’d like the authors to address the following queries:

Introduction

• The authors should consider adding a reference to the sentence “Its undertreatment has been associated with negative effects on the mother and her developing foetus” (see Mitchell & Goodman, Arch Womens Ment Health, 2018)

• When referring to the studies that failed in identifying an association between depressive symptoms and neonatal outcomes the authors might consider to discuss the role of symptoms severity, comorbid anxiety and medications (see Ossola et al., J Matern Fetal Neonatal Med, 2021). This seems particularly relevant as it is not clear whether (or in which percentage) the sample was treated.

• When stating the aims the authors conclude that “if antenatal depression is associated with poor birth and neonatal outcomes, it would emphasize the importance of screening for it as part of routine prenatal care”. This however is only partly descriptive of the study design as the authors also consider post-partum depressive symptoms as a possible maternal outcome. This should be clarified early on.

Methods and Results

• I’m not sure I fully understand the meaning of “a medical doctor attending to severe chronic conditions monthly”. Do the authors mean that an obstetrician-gynaecologists was available only once a month for the most severe patients? Was the psychiatrist that administered the SCID-IV similarly available?

• The authors detailed the power analysis as follow “Assuming a 39% prevalence of antenatal depression, we calculated that a sample of 366 women was needed to achieve a 5% precision at 95% confidence intervals”. I’m not sure I understand based on which aim this sample size was calculated.

• When defining the data collection procedures would be helpful to detail which information were collected at baseline (see Table 1) and how these were dichotomised. As noted above the authors should justify why they dichotomised the following variables: maternal age, maternal BMI, birth weight at delivery, gestational age, Apgar score, postnatal EPDS. Whereas this is useful to represent the data, doing so they are likely to lose nearly a third of the information (see Altman & Royston, BMJ 2006).

• Out of the n=84 women, how many women were depressed in the second and in the third trimester?

• Would also be interesting to know how many women were diagnosed with minor and major depression as these have different risk factors (see Marchesi, Bertoni & Maggini, Obstet Gynecol, 2009). Did the authors found the same results when splitting the predictor in these two groups?

• When defining the confounding variables is not clear how the authors selected eight variables out of the seventeen available as covariates. A reference would be needed here. Especially considering that they excluded alcohol use from the covariates but in the discussion they state that “increased risk behaviours such as alcohol use and smoking which may ultimately affect the foetal intrauterine growth”. Which of the available variables in Table 1 were associated with the selected outcomes in Table 2?

• Following the previous comment, did the authors explore the possible interactions between the predictors? For example does the alcohol mediate the association between antenatal depression and low birth weight? Does any of the known risk/protective factors associated with post-natal depression interact in the association between antenatal depression and the EPDS score?

• It’s not clear to me why they used point-biserial correlations “to measure the strength and direction of the association between antenatal depression and continuous outcomes such as infant’s weight and height at six weeks postdelivery” but not with gestational age and weight at birth. Would also be interesting at least to know whether they found an association between antenatal depression and birth weight and gestational age as continuous variables. Also what’s the difference between point-biserial correlations coefficients and the beta in a linear regression?

• Including so many predictors as in the multivariate logistic regression might results in multi-collinearity issues. Did the authors test for the association among the predictors? Especially when dichotomising the covariates it is possible that some subgroups have in fact zero subjects (e.g., in a 2 by 2 table how many women have chronic illnesses diagnosed during the pregnancy but no chronic illnesses before the pregnancy?). I’m asking because this might affect the p-value.

• I think I don’t understand the meaning of the sentence “The low uptake of exclusive breastfeeding could be due to cultural barriers that young mothers face when trying to implement this”. Could the authors pleas spell out this part of the discussion?

Minor points

• In table 4 the asterisks are missing for the association between HIV status and assisted delivery and infant illness at 6 weeks. Also “predictable” might be changed into “predictors”.

• I’m not a native English speaker but I spotted a couple of typos (e.g. “Maternal postnatal depressive symptoms at six weeks postdelivery post-delivery” and “recorded in the baby’s medical booklet Research team members checked”). Also the last paragraph of the “Data collection procedure” seems redundant with the methods section. The authors should probably proofread the manuscript.

6. PLOS authors have the option to publish the peer review history of their article (what does this mean?). If published, this will include your full peer review and any attached files.

Reviewer #1: No

Reviewer #2: **Yes: **Paolo Ossola

---

## [Author Response · Author response to Decision Letter 0]

16 Feb 2022

RE. Responses to reviewer

Thank you very much for reviewing our manuscript. The comments and suggested by the reviewers have been helpful. Please find below our responses to their comments.

Reviewer #1: This is a very interesting paper aiming to examine the prevalence of antenatal depression and its associations with birth and neonatal outcomes in Zimbabwe. The manuscript is clear, well written and discusses about an important topic. However, there are several pieces of information missing and several points that require improvement, as described below:

Response: Thank you very much for this observation. We have attempted to add the missing information as well as address your comments.

Comment 1: Abstract:

Please make it already very clear in the abstract what are the main aims of the study.

Thank you very much for this observation. Please note that we have clarified this, and the abstract’s introduction now reads: “Antenatal depression is highly prevalent and is associated with negative birth and neonatal outcomes. However, the mechanisms and causality behind these associations remain poorly understood as they are varied. Given the variability in whether associations are present, there is need to have context-specific data to understand the complex causal factors that go into these associations. This study aimed to assess the associations between antenatal depression and birth and neonatal outcomes among women attending maternity care in Harare, Zimbabwe.”

Comment 2: Introduction

Comment 2a: Throughout the text, there are some typos and grammar mistakes. Please proofread and review the paper carefully.

Thank you for this observation. Please note that we have proofread the manuscript.

Comment 2b: The introduction is very well-written, but I am missing a more thorough review of the studies conducted in Africa, but specifically in Zimbabwe.

Thank you for this comment. Please note that to date, there are no studies in Zimbabwe that assessed the associations between antenatal depression and birth/neonatal outcomes in spite high prevalence of antenatal depression (M. Kaiyo-Utete & T. Magwali, 2020; Nyamukoho et al., 2019).

Comment 2c: Also, some citations are missing (e.g. “Its under treatment has been associated with negative effects on the mother and her developing foetus.“;“ paucity of evidence on the associations of antenatal depression in the Zimbabwean context, which has a high prevalence of antenatal depression“). The reader has no clue if certain statements refer to the African or to the global context or if they are based on assumptions or evidence. Please cite accordingly.

Thank you for this observation. Please note that the references have been added.

“Its undertreatment has been associated with negative effects on the mother and her developing foetus (Mitchell & Goodman, 2018).”

“paucity of evidence on the associations of antenatal depression in the Zimbabwean context, which has a high prevalence of antenatal depression (M. Kaiyo-Utete & T. Magwali, 2020; Nyamukoho et al., 2019),”

Comment 3: Methods

Comment 3a: Was the validation of the PHQ-9 and EPDS done with this same population or just in the same health centres?

Thank you for this comment. The validation of the PHQ-9 and EPDS was done with this same population. We used the baseline data to validate the screening tools.

Comment 3b: Please describe in detail how the randomization process for the maternity clinics was conducted.

Thank you for this comment. We have described the randomization process for the maternity clinics; please see in text.

“There are 12 polyclinics under the Harare City Council Directorate. The researcher wrote each clinic’s name on a piece of paper and placed the 12 pieces of paper in a basket. She asked a colleague to randomly pick two pieces of paper out of the basket. The clinics picked were thus selected as study sites for data collection”.

Comment 3c: Did the participants provide written consent? If yes, please clearly state it.

The participants provided a written informed consent. We have stated this in the manuscript.“Written informed consent was obtained from the prospective participants. Pregnant women aged ≤16 years were considered emancipated adults; therefore, no parental consent was obtained.”

Comment 3d: Please describe in details the power analysis conducted to decide on the number of participants needed.

Thank you for this observation. The sample size was calculated based on the validation phase. The main aim of the study was to validate depression screening tools and the assessment of the association between antenatal depression and birth/neonatal outcomes was secondary.

Comment 3e: The prenatal depression diagnostics was done with the SCID interview. How did you account for physical symptoms relevant to the depression diagnosis which are normal in pregnancy?

Thank you very much for this enquiry. During data collection, the study psychiatrists who were administering the SCID-IV had to probe about the participant’s symptoms. For example, if the stated that they were experiencing changes in appetite, sleep and concentration, the psychiatrist further asked when these symptoms started and if the patient felt they were directly caused by the pregnancy. Of note was the sleeping pattern, the psychiatrist asked questions like: “Do you feel you cannot sleep properly because you cannot find a comfortable sleeping position?”

Comment 3f: What is the relevance of the variable “initiated breastfeed within 1h post-delivery” for the research question? How is this associated with prenatal depression given that many times, depending on the maternity care centre procedures, this initiation or lack of, is not a maternal choice?

Thank you for this comment. We included the variable “initiated breastfeeding within 1hr post-delivery” because, according to World Health Organization, new mothers are encouraged to initiate breastfeeding within one hour post-delivery to guarantee that the new-born infant receives colostrum which contains antibodies that protect the infant from infections. We appreciate that in most cases this is not a maternal choice, but it has been found that depressed pregnant women are less likely to initiate breastfeeding early (Abdul Raheem et al., 2019; Eastwood et al., 2017; Figueiredo et al., 2014)or to exclusively breastfeed (Cato et al., 2019; Dias & Figueiredo, 2015). These factors have a bearing in the growth and development of the baby.

Comment 3g: Did the authors account for current pregnancy complications or only for past complications? If not current, why not?

We did not account for current pregnancy complications since we did not collect this data. Pregnant women who presented at the polyclinics and had current complications were referred to a tertiary hospital for further management. The women will therefore not be seen again at these clinics. 

Comment 3h: Please cite the studies supporting the choice for the confounding variables. I am unsure if they belong to the global or regional level.

Thank you. Please see in text the references.

Comment 3i: Authors mentioned that participants were recorded as depressed or non-depressed. But from those scored as depressed, what is the percentage/proportion of minor and major depression? Please add this to the text.

Thank you for this comment. We have added this in text.

Comment 3j: Authors mentioned that the depressed participants were referred to the mental health nurse or psychotherapy. When did participants started with this treatment? What kind of treatment was given? How has this impacted mental health measurements postpartum? You need to adjust for this important variable in the analysis.

Thank you for this observation. When the study psychiatrist made a diagnosis of depression after administering the SCID-IV, the participant was given a referral letter to the mental health nurse of for psychotherapy. The research assistants directed the participants to the nurse in charge at the clinic who would then take the participant for treatment as suggested by the study psychiatrist. Unfortunately, we did not follow up to assess whether the participant received the recommended treatment. We do appreciate this in the discussion and state it as one of our study limitations.

Although the pregnant women who had SCID-diagnosed minor/major depression were referred to the Friendship Bench or mental health nurse for further management, information on management outcomes was not available; the fact that women may have received treatment for antenatal depression could explain some of the lack of association between antenatal depression and many birth and neonatal outcomes.

Comment 4: Results

Comment 4a: Were there any differences between the participants who remained until the end of the study and the N=21 who were lost in the follow up?

There were no differences between those that remained until the end of the study and those lost to follow up.

Comment 4b: Participants can be either primipara or multiparas, not both. Please correct it in the text.

Thank you for this observation. This has been corrected.

Comment 4c: Table1: the authors mentioned that covariates were chosen based on previous literature and evidence, but aren’t religion and social support quite important in the regional context? 

Thank you for this observation. The covariates were chosen based on previous literature and evidence. Religion and social support are important factors associated with antenatal depression (Kaiyo-Utete et al, 2020; Nyamukoho et al, 2019); however, there is paucity of evidence that supports their effects on birth and neonatal outcomes, hence they were not included among the covariates.

Also, were the randomly selected clinics settled in a rural or urban region? This should be acknowledged, given that if they are in an urban setting, women attending it might not be comparable to those in a rural setting with less resources to attend and pay for the maternity care.

Thank you for this observation. The clinics are in urban areas. We have highlighted this in the methodology section.

Comment 4d: Please state the percentage of participants diagnosed with minor and major depression in the sample in Table 1.

Thank you for this observation. 29/84 (34.5%) of the depressed had major depression compared to 55/84 (65.5%) who had major depression.

Comment 4e: Although the authors clearly described depression levels and infant characteristics in the postpartum phase, it would be interesting to see a table with the outcomes from babies whose mothers were antenatally depressed. It is unclear, especially considering prenatally depressed mothers were offered and potentially received treatment during pregnancy, how many of those were still depressed in the postpartum and the potential effects/improvement on baby outcomes for those receiving therapy.

Thank you for this comment. Please not that the depression levels described in this manuscript are for pregnant women. The outcomes, though, include for both depressed and non-depressed pregnant women to assess associations with antenatal depression. Depression in pregnancy was assessed using the SCID-IV while depressive symptoms in the postpartum period were assessed using the EPDS at 6 weeks postnatal clinic visit.

Comment 4f: Table 2: It would be important to add the percentage of babies not being breastfed at all (e.g. only formula feeding). 

Thank you for this comment. Formula feeding is not very common in the sample due to the financial costs attached to formula feeding hence the reason we categorised it as non-exclusive breastfeeding.

Comment 5: Discussion

Comment 5a: In the second paragraph, authors justify the association between antenatal depression and LBW risk behaviours. However, it is important to contextualize it to your sample. Based on Table 1, the participants have a low percentage of risk behaviours. Maybe you could justify it if the depressed moms have a higher rate of risky behaviours upon comparison with the non-depressed one. But this needs to be described in text or tables.

In our study, there was no significant association between birth weight and LBW; this could because of the bias introduced in the study due to face-to-face interviews which might have influenced the participants to give socially desirable answers.

Comment 5b: Authors also mentioned cultural barriers associated with breastfeeding. What are those in Zimbabwe? It would be important to contextualize it in the region where the study was conducted as well as with the sample here studied. 

Thank you very much for this comment. Please see in the text, we have added references that contextualize the cultural barriers associated with breastfeeding in Zimbabwe (Muchacha & Mtetwa, 2015; Mundagowa et al., 2019; Nduna T, 2015; Yaya et al., 2020).

Comment 5c: Please make sure to make it clear that the association between prenatal depression and exclusive breast was quite small and how relevant is it.

Thank you for this comment. Please note that we have addressed it in the text.

Comment 5d: Did the authors considered that the lack of statistical significance between depression and certain outcomes may be due to the fact that most participants were diagnosed with minor depression? Since this is not clearly described in text, the reader has no clue if this is a possibility. Please make it clear.

Thank you for this comment. We have added the proportions of minor and major depression among the participants and this consideration.

Reviewer #2: 

In this original article, the Authors aim to determine the association between antenatal depression during the second or third trimester and birth and neonatal outcomes assessed at 6 weeks post-partum in a cohort of 354 Zimbabwean women. They found that a diagnosis of major and minor antenatal depression was associated with low birth weight in the new-borns, with a lower rate of exclusive breastfeeding and with higher rate of depressive symptoms beyond the EPDS cut-off.

The study is interesting because to date only few prospective studies evaluated the impact of prenatal depression in sub-Saharian countries. Some methodological issues, hovewer, lower my enthusiasm for the manuscript. Specifically the authors considered the antenatal depression as the main predictor of ten different outcomes. In a second step of the analyses they controlled for eight different covariates. Moreover they dichotomised most of the variable with the risk of losing some important information.

If possible I’d like the authors to address the following queries:

Introduction

• The authors should consider adding a reference to the sentence “Its undertreatment has been associated with negative effects on the mother and her developing foetus” (see Mitchell & Goodman, Arch Womens Ment Health, 2018)

Thank you for this. We have added the reference as per your suggestion.

• When referring to the studies that failed in identifying an association between depressive symptoms and neonatal outcomes the authors might consider to discuss the role of symptoms severity, comorbid anxiety and medications (see Ossola et al., J Matern Fetal Neonatal Med, 2021). This seems particularly relevant as it is not clear whether (or in which percentage) the sample was treated.

Thank you. We have noted that in our discussion.

• When stating the aims the authors conclude that “if antenatal depression is associated with poor birth and neonatal outcomes, it would emphasize the importance of screening for it as part of routine prenatal care”. This however is only partly descriptive of the study design as the authors also consider post-partum depressive symptoms as a possible maternal outcome. This should be clarified early on.

Thank you. We have clarified. It now reads: “If antenatal depression is associated with poor birth and neonatal outcomes and postnatal depressive symptoms, it would emphasize the importance of screening for it as part of routine prenatal care.” 

Methods and Results

• I’m not sure I fully understand the meaning of “a medical doctor attending to severe chronic conditions monthly”. Do the authors mean that an obstetrician-gynaecologists was available only once a month for the most severe patients? Was the psychiatrist that administered the SCID-IV similarly available?

Thank you for this comment. 

The clinics do not have a resident medical practitioner at all; they are staffed by midwives and general nurses. In cases of obstetric complications, pregnant women are referred to tertiary hospital for further management. However, once a month, a medical practitioner (not a specialist) visits the clinic to attend to patients with chronic illnesses such as hypertension, diabetes mellitus and heart conditions. The psychiatrists who administered the SCID were part of the research team, they were not stationed at these clinics.

• The authors detailed the power analysis as follow “Assuming a 39% prevalence of antenatal depression, we calculated that a sample of 366 women was needed to achieve a 5% precision at 95% confidence intervals”. I’m not sure I understand based on which aim this sample size was calculated.

Thank you for this observation. The sample size was calculated based on the validation phase. The main aim of the study was to validate depression screening tools and the assessment of the association between antenatal depression and birth/neonatal outcomes was secondary.

• When defining the data collection procedures would be helpful to detail which information were collected at baseline (see Table 1) and how these were dichotomised. As noted above the authors should justify why they dichotomised the following variables: maternal age, maternal BMI, birth weight at delivery, gestational age, Apgar score, postnatal EPDS. Whereas this is useful to represent the data, doing so they are likely to lose nearly a third of the information (see Altman & Royston, BMJ 2006).

Thank you for this comment. We dichotomised the variables to make it easier to represent the data. However, when we ran the point-biserial correlation with these variables as continuous variables there were no differences in the associations, hence we did not change the analysis in the manuscript.

• Out of the n=84 women, how many women were depressed in the second and in the third trimester?

Thank you. We did not analyse depression status per trimester because the women were recruited either in the second trimester or third trimester. 

• Would also be interesting to know how many women were diagnosed with minor and major depression as these have different risk factors (see Marchesi, Bertoni & Maggini, Obstet Gynecol, 2009). 

Did the authors found the same results when splitting the predictor in these two groups?

Thank you for this observation. 29/84 (34.5%) of the depressed had major depression compared to 55/84 (65.5%) who had major depression. For this paper, we did not split the data into minor and major depression. The study was more of an exploratory study to assess whether there are any associations between antenatal depression and birth/neonatal outcomes.

• When defining the confounding variables is not clear how the authors selected eight variables out of the seventeen available as covariates. A reference would be needed here. Especially considering that they excluded alcohol use from the covariates but in the discussion, they state that “increased risk behaviours such as alcohol use and smoking which may ultimately affect the foetal intrauterine growth”. Which of the available variables in Table 1 were associated with the selected outcomes in Table 2?

Thank you for this comment. Please see in text; we have added the references on the chosen covariates. 

• Following the previous comment, did the authors explore the possible interactions between the predictors? For example, does the alcohol mediate the association between antenatal depression and low birth weight? Does any of the known risk/protective factors associated with post-natal depression interact in the association between antenatal depression and the EPDS score?

There were no significant relationships between predictors as we tested for collineality thus we ruled out multicollineality.

• It’s not clear to me why they used point-biserial correlations “to measure the strength and direction of the association between antenatal depression and continuous outcomes such as infant’s weight and height at six weeks postdelivery” but not with gestational age and weight at birth. Would also be interesting at least to know whether they found an association between antenatal depression and birth weight and gestational age as continuous variables. Also what’s the difference between point-biserial correlations coefficients and the beta in a linear regression?

Thank you for this comment. We ran the point-biserial correlations to measure the strength and direction between antenatal depression and 

a. Gestational age at birth (rpb=0.024, p=0.65)

b. Birthweight (rpb=0.096, p=0.07)

c. Apgar score at 5 minutes (rpb=0.049, p=0.361)

Just like with the logistic regression, the point-biserial correlation showed that there was a borderline association between antenatal depression and birthweight as a continuous variable but there were associations between gestational age and Apgar score.

The point-biserial correlation was used we were assessing associations between a binary variable (antenatal depression) and a continuous variable (infant’s weight and height at 6 weeks postdelivery) whereas a linear regression assesses associations between continuous variables.

• Including so many predictors as in the multivariate logistic regression might results in multi-collinearity issues. Did the authors test for the association among the predictors? Especially when dichotomising the covariates it is possible that some subgroups have in fact zero subjects (e.g., in a 2 by 2 table how many women have chronic illnesses diagnosed during the pregnancy but no chronic illnesses before the pregnancy?). I’m asking because this might affect the p-value.

We tested pairwise relationship and they were not correlated. There were no groups with zero subjects. But the changes in the p-values are purely due to additive effects.

• I think I don’t understand the meaning of the sentence “The low uptake of exclusive breastfeeding could be due to cultural barriers that young mothers face when trying to implement this”. Could the authors pleas spell out this part of the discussion?

The sentence “The low uptake of exclusive breastfeeding could be due to cultural barriers that young mothers face when trying to implement this” means that not many mothers are exclusively breastfeeding their babies. The reasons for this could be due to cultural barriers, such as the young mother being encouraged to supplement the baby’s feeding by the elderly women in the family. We have spelled this out in the discussion.

Minor points

• In table 4 the asterisks are missing for the association between HIV status and assisted delivery and infant illness at 6 weeks. Also “predictable” might be changed into “predictors”.

Thank you for this observation. Please note that we have put the asterisks in Table 4. 

• I’m not a native English speaker but I spotted a couple of typos (e.g. “Maternal postnatal depressive symptoms at six weeks postdelivery post-delivery” and “recorded in the baby’s medical booklet Research team members checked”). Also the last paragraph of the “Data collection procedure” seems redundant with the methods section. The authors should probably proofread the manuscript.

Thank you so much for this comment. We have attended to the typos.

---

## [Decision Letter · Decision Letter 1]

26 Apr 2022

PONE-D-21-18898R1Antenatal depression: Associations with birth and neonatal outcomes among women attending maternity care in Harare, ZimbabwePLOS ONE

Dear Dr. Kaiyo-Utete,

Thank you for submitting your manuscript to PLOS ONE. After careful consideration, we feel that it has merit but does not fully meet PLOS ONE’s publication criteria as it currently stands. Therefore, we invite you to submit a revised version of the manuscript that addresses the points raised during the review process.

The authors have made several substantial changes to their manuscript. However, a new reviewer has suggested several areas for further clarification, especially regarding explanation of terms/concepts and limitations to the study design. In addition, paying attention to using causal language, only where the study does show causality; in all other cases, correlations/associations should be reported.

The reviewer recommends some grammar/style changes, such as not starting a sentence with "However". These types of comments can be changed or not at the authors' desire. 

We look forward to receiving your revised manuscript.

Kind regards,

Michael Wells

Academic Editor

PLOS ONE

Reviewers' comments:

Reviewer's Responses to Questions

**Comments to the Author**

1. If the authors have adequately addressed your comments raised in a previous round of review and you feel that this manuscript is now acceptable for publication, you may indicate that here to bypass the “Comments to the Author” section, enter your conflict of interest statement in the “Confidential to Editor” section, and submit your "Accept" recommendation.

Reviewer #3: (No Response)

2. Is the manuscript technically sound, and do the data support the conclusions?

Reviewer #3: Partly

3. Has the statistical analysis been performed appropriately and rigorously? 

Reviewer #3: Yes

4. Have the authors made all data underlying the findings in their manuscript fully available?

Reviewer #3: No

5. Is the manuscript presented in an intelligible fashion and written in standard English?

Reviewer #3: Yes

6. Review Comments to the Author

Reviewer #3: Thank you for the opportunity to review this important manuscript. This study aimed to assess the associations between antenatal depression and birth and neonatal outcomes among women attending maternity care in Harare, Zimbabwe- a location where this type of assessment has not been done prior. While I think the authors made some adjustments based on prior reviewers, there are still major revisions that need to be made before this would be considered publishable.

1. Abstract: this study looks at associations. I would rephrase/remove talk of causality in the introduction. Few studies have been done in Zimbabwe, which is a fine justification for the study.

2. Citation 20 needs full brackets in the text. It is missing

3. Introduction: do not start a paragraph with “However”—suggest rewriting second and third paragraphs to help with flow. Very much like the last paragraph in the introduction setting up the study- just need a bit of a cleaner intro and segue into that study purpose paragraph.

4. Exposure: You say “SCID-IV is administered by trained mental health professionals” and then a few sentences later “Here, trained study psychiatrists administered the SCID-IV during data collection.” You do not need both.

5. Outcomes-birth outcomes: Is medical booklets correct term? This isn’t used in the USA. Please elaborate.

6. Missing a period in the sentence here “Infant’s weight and height were measured routinely by the attending midwife or nurse and recorded in the baby’s medical booklet [ADD PERIOD] Research team members checked the…

7. If this was part of a validation study. Why is Pre and postnatal depression measured using different tools?

8. Why was prior perinatal depression not included as a covariate. It is the MOST predictive factor for future perinatal depression.

9. All statistical analysis was done in STATA version 14[ADD PERIOD]

10. Be consistent with how numbers are formatted.

11. Table 1. What does “Negative life event in the past one year” mean? How was it measured?

12. Please put the antenatal depression findings in table 1. IT will be easier for readers.

13. Please add unit to weight.

14. Table 3. Remove statistical significance * from birthweight and remove the 1’s from the comparison group under Unadjusted OR

15. Table 4 is really hard to read. Why are all outcomes included even though most were not associated with the outcome in the unadjusted model? I would suggest using a 0.2 cutoff and only showing birthweight, breastfeeding at 6 weeks, and postnatal depression.

16. Agree with prior reviewer- please include tables with demographics and outcomes from babies whose mothers were antenatally depressed vs. those not. The frequencies and counts would be valuable. Differences in demos would also be valuable to assess.

16. Are there differences in the 2 clinics? I would suggest added the clinic to the model in case there are.

17. Discussion: “One possible explanation for this association is that mothers with depression have reduced cognitive function, which may lead to poor maternal nutrition, poor health seeking behaviours and increased risk behaviours such as alcohol use and smoking which may ultimately affect the

foetal intrauterine growth.”—this is a really big and bold statement with a lot of maternal blame. Could there not be other reasons for low birth weight in a developing country?

18. Discussion: It is stated that screening tools overestimate the prevalence of depression but they used a screening tool for postnatal depression.

19. Conclusion: Universal screening for antenatal depression is recommended by the American College of Gynecologist. May need to be added to WHO’s recommendations.

20. From prior reviewer—please correct parity description in the tables- Participants can be either primipara or multiparas

21. If the authors believe the lack of statistical significance between depression and certain outcomes may be due to the fact that most participants were diagnosed with minor depression—assess this. You have the data!

22. The fact that depression was only measured once during pregnancy is a major limitation.

23. IN the response to reviewers it is stated that multicollinearity was assessed. Please state that in the methods and results.

7. PLOS authors have the option to publish the peer review history of their article (what does this mean?). If published, this will include your full peer review and any attached files.

Reviewer #3: No

---

## [Author Response · Author response to Decision Letter 1]

7 Jun 2022

1. Abstract: this study looks at associations. I would rephrase/remove talk of causality in the introduction. Few studies have been done in Zimbabwe, which is a fine justification for the study.

Thank you very much for this comment. Please note that we have revised the abstract’s introduction by removing the phrase that refers to causality in the introduction. The statement now reads: Given the variability in whether associations are present, there is need to have context-specific data to understand the complex factors that go into these associations.

2. Citation 20 needs full brackets in the text. It is missing

Thank you very much for this observation. Please note that we have added the missing brackets.

Postnatal depression is often a sequela of untreated antenatal depression, with almost 50% of women who are diagnosed with postnatal depression having had signs of depression in pregnancy [20].

3. Introduction: do not start a paragraph with “However”—suggest rewriting second and third paragraphs to help with flow. Very much like the last paragraph in the introduction setting up the study- just need a bit of a cleaner intro and segue into that study purpose paragraph.

Thank you very much for this observation. Please note that we have rewritten the beginning sentence of the second paragraph as follows: Some studies, however, have shown that there are no statistically significant associations between antenatal depression and adverse birth and neonatal outcomes [23-26].

4. Exposure: You say “SCID-IV is administered by trained mental health professionals” and then a few sentences later “Here, trained study psychiatrists administered the SCID-IV during data collection.” You do not need both.

Thank you very much for the above observation. Please note that we felt that including the statement “Here, trained study psychiatrists administered the SCID-IV during data collection.” would show the specific trained mental health professionals we used to collect the data. However, we have removed the statement from the revised manuscript.

5. Outcomes-birth outcomes: Is medical booklets correct term? This isn’t used in the USA. Please elaborate.

Thank you very much for this comment. The term “medical booklet” refers to a book that has the infant’s medical records from delivery. 

6. Missing a period in the sentence here “Infant’s weight and height were measured routinely by the attending midwife or nurse and recorded in the baby’s medical booklet [ADD PERIOD] Research team members checked the…

Thank you very much for this observation. Please note that this has been addressed. 

Infant’s weight and height were measured routinely by the attending midwife or nurse and recorded in the baby’s medical booklet. Research team members checked the infant’s medical records to assess whether the infant had had a clinic visit for diarrhoea, respiratory infection, or any other illness in the previous six weeks.

7. If this was part of a validation study. Why is Pre and postnatal depression measured using different tools?

The study validated the translated version of the EPDS among pregnant women. We used the SCID-IV as the diagnostic tool to assess pre/antenatal depression as this tool gave a definitive diagnosis. This paper addressed antenatal depression, not the depressive symptoms.

8. Why was prior perinatal depression not included as a covariate. It is the MOST predictive factor for future perinatal depression.

Thank you very much for this comment.

We appreciate that previous perinatal depression is the MOST predictive factor for future perinatal depression. However, perinatal depression is not assessed routinely in Zimbabwean antenatal care services so the women will not have the history of prior perinatal depression.

9. All statistical analysis was done in STATA version 14[ADD PERIOD]

Thank you very much for this observation. Please note that we have added the period. “All statistical analysis was done in STATA version 14 (2015)”.

10. Be consistent with how numbers are formatted.

Thank you very much for this comment. Please note that we have addressed this comment.

11. Table 1. What does “Negative life event in the past one year” mean? How was it measured?

“Negative life event in the past year” means any event in the pregnant woman’s life that might be deemed a significant loss to her such as divorce or separation from husband, death of a significant or loved one, loss of a source of income, being evicted from home. 

The pregnant women were asked five questions whether they had lost a love/significant one to death, home, job, if they had divorced or separated from their husband. If they answered Yes to any of the questions, they were deemed to have had a negative life event.

12. Please put the antenatal depression findings in table 1. IT will be easier for readers.

Thank you very much for this comment. Please note that this has been addressed in Table 1.

13. Please add unit to weight.

Thank you for this observation. Please note that this has been addressed.

14. Table 3. Remove statistical significance * from birthweight and remove the 1’s from the comparison group under Unadjusted OR

Thank you for this observation. Please note that this has been addressed in Table 3.

15. Table 4 is really hard to read. Why are all outcomes included even though most were not associated with the outcome in the unadjusted model? I would suggest using a 0.2 cutoff and only showing birthweight, breastfeeding at 6 weeks, and postnatal depression.

Thank you very much for this observation. Please note that we have corrected the table as recommended.

16. Agree with prior reviewer- please include tables with demographics and outcomes from babies whose mothers were antenatally depressed vs. those not. The frequencies and counts would be valuable. Differences in demos would also be valuable to assess.

Thank you for this comment. Please note that we have addressed this (see Table 2).

17. Are there differences in the 2 clinics? I would suggest added the clinic to the model in case there are.

Thank you very much for this comment. There were no differences in the two clinics.

18. Discussion: “One possible explanation for this association is that mothers with depression have reduced cognitive function, which may lead to poor maternal nutrition, poor health seeking behaviours and increased risk behaviours such as alcohol use and smoking which may ultimately affect the foetal intrauterine growth.”—this is a really big and bold statement with a lot of maternal blame. Could there not be other reasons for low birth weight in a developing country?

Thank you very much for this observation. Please note that we appreciate that reduced maternal cognitive function is not the only reason that may lead to low birth weight that is why we have stated it as “One possible explanation for this association…”. This just confirms that there is a complex relationship among the factors associated with birth weight.

19. Discussion: It is stated that screening tools overestimate the prevalence of depression, but they used a screening tool for postnatal depression.

Thank you for this comment. Since the objective of the study was to assess the associations between antenatal depression and birth and neonatal outcomes, we needed a definitive diagnosis for antenatal depression which the SCID-IV provided. We also took into cognizant that the Shona-EPDS was already validated in the study population.

20. Conclusion: Universal screening for antenatal depression is recommended by the American College of Gynecologist. May need to be added to WHO’s recommendations.

Thank you for this comment. Please note that screening for depression is not yet part of antenatal care services in Zimbabwe hence the recommendation. We have however appreciated that universal screening is recommended by American College of Gynecologists in our recommendations. The statement now reads: 

“Accordingly, there is great need for more research to identify whether screening for and subsequently treating antenatal depression can improve these outcomes; this data is needed to support universal screening for antenatal depression which is recommended by the American College of Gynecologists.”

21. From prior reviewer—please correct parity description in the tables- Participants can be either primipara or multiparas

Thank you very much for this correction. We appreciate this, however, we grouped the participants as either nullipara (those that were in their first pregnancy and had never given birth before) and primipara or above (these had given birth once or more times prior to this study).

22. If the authors believe the lack of statistical significance between depression and certain outcomes may be due to the fact that most participants were diagnosed with minor depression—assess this. You have the data!

Thank you for this comment. Please note that we have retracted that statement The study was more of an exploratory study to assess whether there are any associations between antenatal depression and birth/neonatal outcomes.

23. The fact that depression was only measured once during pregnancy is a major limitation.

Thank you very much for this comment. Please note that we acknowledge this limitation for our study: “Firstly, antenatal depression was assessed once, the participant might have not been depressed at the time of data collection but later developed depression or vice versa”

24. IN the response to reviewers, it is stated that multicollinearity was assessed. Please state that in the methods and results.

Thank you very much for this comment. Please note that we have included this as recommended.

---

## [Editor Report · Decision Letter 2]

21 Jun 2022

Antenatal depression: Associations with birth and neonatal outcomes among women attending maternity care in Harare, Zimbabwe

PONE-D-21-18898R2

Dear Dr. Kaiyo-Utete,

We’re pleased to inform you that your manuscript has been judged scientifically suitable for publication and will be formally accepted for publication once it meets all outstanding technical requirements.

Kind regards,

Michael Wells

Academic Editor

PLOS ONE
---

## [Editor Report · Acceptance letter]

25 Aug 2022

PONE-D-21-18898R2 

Antenatal depression: Associations with birth and neonatal outcomes among women attending maternity care in Harare, Zimbabwe 

Dear Dr. Kaiyo-Utete:

I'm pleased to inform you that your manuscript has been deemed suitable for publication in PLOS ONE. Congratulations! Your manuscript is now with our production department. 

Kind regards, 

on behalf of

Dr. Michael Wells 

Academic Editor

PLOS ONE